

# Enhancing reliable and energy-efficient UAV communications with RIS and deep reinforcement learning

Wasim Ahmad[1], Umar Islam[2], Abdulkadhem A. Abdulkadhem[3], Babar Shah[4], Fernando Moreira[5] and Ali Abbas[6]

[1] School of Arts and Creative Technology University of Greater Manchester, Bolton, United Kingdom
[2] Department of Computer Science, IQRA National university, Peshawar, Swat Campus, Pakistan
[3] Department of Cyber Security, College of Sciences, Al-Mustaqbal University, Hillah, Babil, Iraq
[4] College of Technological Innovation, Zayed University, Dubai, United Arab Emirates
[5] REMIT, IJP, Universidade Portucalense, Porto, Portugal
[6] Middle East College, Muscat, Oman

Corresponding authors
Babar Shah, Babar.Shah@zu.ac.ae
Fernando Moreira, fmoreira@upt.pt

## ABSTRACT

The rapid growth in wireless communication demands has led to a surge in research on technologies capable of enhancing communication reliability, coverage, and energy efficiency. Among these, uncrewed aerial vehicles (UAV) and reconfigurable intelligent surfaces (RIS) have emerged as promising solutions. Prior research on using deep reinforcement learning (DRL) to integrate RIS with UAV concentrated on enhancing signal quality and coverage, but it ignored the challenges caused by electromagnetic interference (EMI). This article introduces a novel framework addressing the challenges posed by EMI from Gallium nitride (GaN) power amplifiers in RIS-assisted UAV communication systems. By integrating DRL with quadrature phase shift keying (QPSK) modulation, the proposed system dynamically optimizes UAV deployment and RIS configurations in real-time, mitigating EMI effects, improving signal-to-interference-plus-noise ratio (SINR), and enhancing energy efficiency. The framework demonstrates superior performance, with an SINR improvement of up to 6.5 dB in interference-prone environments, while achieving a 38% increase in energy efficiency compared to baseline models. Additionally, the system significantly reduces EMI impact, with a mitigation rate of over 70%, and extends coverage area by 35%. The integration of QPSK and DRL allows for real-time decision-making that balances communication quality and energy consumption. These results show the system's potential to outperform traditional methods, particularly in dynamic and challenging environments such as urban, disaster recovery, and remote settings.

# INTRODUCTION

The rapid growth in wireless communication demands has led to a surge in research on technologies capable of enhancing communication reliability, coverage, and energy efficiency (*Mohsan et al., 2022*; *Aldaej, Ahanger & Ullah, 2023*; *Alkanhel et al., 2023*). Among these, uncrewed aerial vehicles (UAVs) and reconfigurable intelligent surfaces

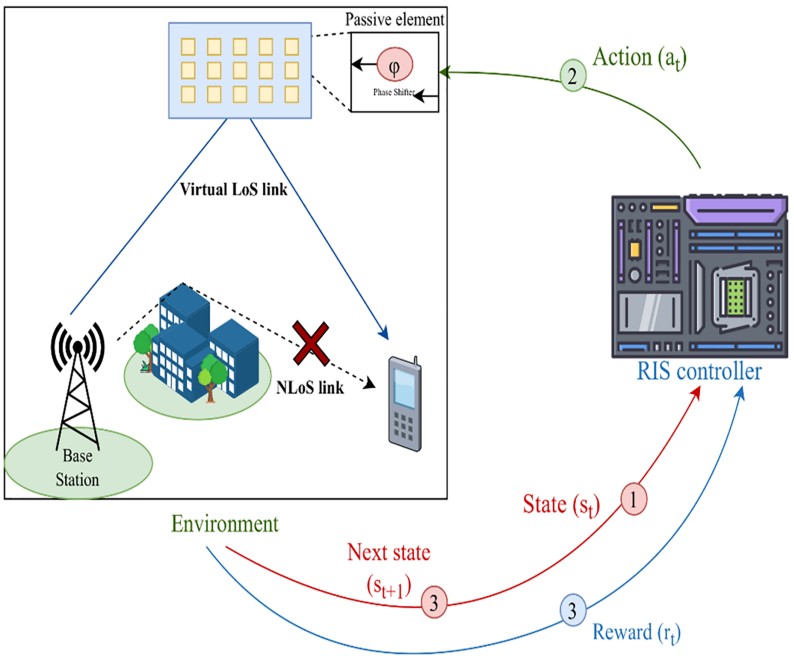

**Figure 1 Previous UAV-RIS assisted system.**

(RIS) have emerged as promising solutions (*Bansal et al., 2023*; *Bithas et al., 2024*; *Emami, 2023*). UAVs offer flexible and mobile platforms that provide line-of-sight communication links, while RIS enables dynamic control over the wireless propagation environment through phase shift adjustments (*Wu et al., 2024*; *Cang et al., 2023*; *You et al., 2023*; *Iqbal et al., 2023*). However, the influence of electromagnetic interference (EMI) generated by high-performance components, such as Gallium nitride (GaN) power amplifiers, on these systems has yet to be fully explored (*Javad-Kalbasi, Al-Abiad & Valaee, 2023*; *Ji et al., 2023*; *Jiao, Xie & Ding, 2022*; *Lahmeri, Kishk & Alouini, 2021*; *Li et al., 2024*).

This article focuses on investigating the influence of GaN power amplifier EMI on the RIS-assisted UAV communication systems. Additionally, we propose a novel framework that integrates deep reinforcement learning (DRL) with UAV-RIS systems to mitigate the adverse effects of EMI while optimizing communication parameters in real-time. By incorporating DRL, the UAVs and RIS can continuously adjust their configurations based on environmental changes, such as varying interference levels, user locations, and obstacles. The concept of UAV-RIS assisted system is shows in Fig. 1.

The motivation behind this research stems from the need to address the challenges posed by EMI in UAV-RIS communication systems, particularly EMI from GaN power amplifiers. These high-power components, often used in amplifying signals, can generate significant electromagnetic interference, adversely impacting wireless communication performance (*Nguyen et al., 2021*, *2022*; *Pogaku et al., 2022*).

The primary objective of this study is to propose an optimized UAV-RIS communication framework that leverages DRL to dynamically adjust UAV deployments and RIS configurations, effectively mitigating the adverse impacts of GaN power amplifier-induced EMI. Unlike conventional approaches that primarily focus on improving coverage and signal strength, this framework actively targets EMI-induced signal degradation, enhancing signal-to-interference-plus-noise ratio (SINR), energy efficiency, and communication reliability in real-time.

The specific objectives of this research are as follows:

(a) To analyze the influence of GaN power amplifier EMI on UAV-RIS-assisted communication systems
- Investigate how high-power GaN amplifiers introduce electromagnetic disturbances, affecting SINR and overall communication reliability in UAV-RIS networks.

(b) To design a DRL-based optimization framework for real-time adaptation of UAV and RIS parameters
- Develop a DRL model that dynamically adjusts UAV positioning and RIS phase shifts based on environmental interference levels to maximize SINR and minimize EMI effects.

(c) To integrate quadrature phase shift keying (QPSK) modulation for enhanced spectral efficiency and interference resilience
- Employ QPSK to improve data throughput and signal robustness in interference-prone environments, supporting stable communication links.

(d) To evaluate the proposed framework under various interference-prone scenarios
- Perform extensive simulations to validate improvements in SINR, energy efficiency, EMI mitigation, and communication reliability compared to baseline UAV-RIS systems.

(e) To demonstrate the scalability and adaptability of the framework in complex communication environments
- Test the model's performance in urban, rural, and disaster recovery scenarios where EMI is prevalent, showcasing its adaptability to real-world conditions.

Although there has been extensive research on UAV and RIS technologies independently, the combined effects of EMI from GaN power amplifiers on UAV-RIS communication systems remain largely unexplored. Previous studies on integrating RIS with UAVs using DRL focused on improving signal quality and coverage but did not account for the challenges posed by EMI. This study addresses this gap by investigating the combined effects of EMI and the mitigation strategies provided by DRL-optimized UAV-RIS systems. This article makes the following contributions:

(a) A novel DRL-based framework that dynamically adapts UAV positions and RIS configurations to mitigate the adverse effects of GaN power amplifier EMI on communication performance.

(b) An in-depth analysis of the influence of EMI on RIS-assisted UAV communication systems, and how DRL can be used to optimize signal quality and energy efficiency in real-time.

(c) Comprehensive simulation results show significant improvements in SINR, energy efficiency, coverage, and latency in EMI-prone environments when using the proposed DRL-based approach.

Despite the promising benefits of RIS and UAVs, dynamic control of RIS reflection patterns and the real-time adjustment of UAV positions in the presence of EMI present several challenges. The proposed DRL framework must operate within physical and positional constraints to ensure stable communication links while optimizing energy efficiency. Furthermore, GaN power amplifiers can introduce significant EMI, necessitating intelligent strategies to mitigate its effects without compromising communication quality. In summary, this article introduces a novel UAV-RIS communication framework that uses DRL to mitigate the effects of GaN power amplifier EMI, improving communication performance in dynamic and challenging environments. By optimizing UAV and RIS configurations in real time, this research paves the way for future developments in resilient and adaptive wireless networks.

## LITERATURE REVIEW

The growing demand for enhanced wireless communication capabilities has led to the exploration of RIS and their integration with UAVs for efficient and reliable communication networks. The concept of RIS-assisted UAV communication has been extensively studied to improve signal strength, reduce interference, and enhance connectivity in various network settings. For instance, *Mohsan et al. (2022)* highlighted current trends and challenges in massive networks, emphasizing the need for efficient resource management strategies in RIS-assisted UAV systems. Similarly, *Nguyen et al. (2021)* proposed a deep reinforcement learning-based approach for efficient resource allocation in multi-UAV networks supported by RIS, demonstrating significant improvements in network performance and resource optimization. Further research by *Ji et al. (2023)* presented a deep reinforcement learning-based optimization for UAV-non-orthogonal multiple access (NOMA) downlink networks, utilizing RIS to enhance communication quality and manage interference. *Puspitasari & Lee (2023)* provided a comprehensive survey on the application of reinforcement learning techniques for RIS in wireless communications, underscoring the importance of intelligent decision-making in dynamic environments. *Rahmatov & Baek (2023)* discussed current research, challenges, and future trends in RIS-carried UAV communication, identifying key areas for improvement, such as scalability and adaptive control mechanisms. In addressing the security and privacy concerns associated with RIS-assisted UAV communication, *Iqbal et al. (2023)* proposed a deep reinforcement learning-based resource allocation framework for secure communication, emphasizing the integration of RIS to enhance data protection. *Nguyen et al. (2022)* explored the potential of RIS-assisted UAV communication for IoT applications, incorporating wireless power transfer and deep

reinforcement learning to optimize energy efficiency and coverage. *Javad-Kalbasi, Al-Abiad & Valaee (2023)* examined energy-efficient communication strategies for RIS-assisted UAV networks, employing genetic algorithms to optimize resource allocation and reduce energy consumption. In addition, *Emami (2023)* investigated the use of deep reinforcement learning for joint cruise control and data acquisition in UAV-assisted sensor networks, demonstrating the capability of RIS to enhance system efficiency. *Bansal et al. (2023)* proposed a RIS selection scheme for UAV-based multi-user networks, addressing challenges related to imperfect and outdated channel state information (CSI). *Pogaku et al. (2022)* and *Taimoor, Ferdouse & Ejaz (2022)* conducted a survey on optimization techniques and performance analysis for UAV-assisted RIS systems, providing valuable insights into the design and implementation of next-generation wireless networks. The deployment of UAV-mounted RIS to enhance mobile edge computing and optimize resource management has also been explored by various researchers. *You et al. (2023)* utilized deep reinforcement learning to optimize UAV-D2D communications in RIS-assisted networks, showing potential improvements in network throughput and latency. *Bithas et al. (2024)* conducted a stochastic analysis of UAV-assisted communications with RIS, focusing on shadowing effects and their impact on network performance. *Nguyen et al. (2021, 2022)* highlighted the role of deep reinforcement learning in optimizing UAV communications with RIS for IoT, underlining the importance of intelligent control and adaptability. *Khan et al. (2024, 2023b)* presented layered computing solutions that can be seamlessly integrated with RIS-assisted UAV architectures to enable efficient offloading of computationally intensive deep learning models. Moreover, recent studies have explored the use of advanced optimization techniques to further enhance UAV-RIS communication systems. *Wu et al. (2024)* employed a double deep Q-network (DDQN) approach for optimizing mobile edge computing systems in UAV-mounted RIS networks. *Alkanhel et al. (2023)* integrated slime mold optimization with deep learning for resource allocation in UAV-enabled wireless networks, showcasing the potential of hybrid approaches to optimize performance. *Cang et al. (2023)* proposed joint deployment and resource management strategies for visible light communication (VLC)-enabled RIS-assisted UAV networks, emphasizing the need for seamless integration of multiple communication technologies. Reinforcement learning-based approaches continue to gain traction in optimizing RIS-assisted UAV networks, with studies such as those by *Wang et al. (2021)* and *Aldaej, Ahanger & Ullah (2023)* focusing on multi-agent reinforcement learning for trajectory planning and blockchain-enabled secure data transmission, respectively. The adoption of deep bidirectional learning for improving outage probability in aerial RIS-assisted communication systems has also been proposed by *Rahman et al. (2024)*, highlighting the role of advanced machine learning techniques in addressing reliability issues. *Lahmeri, Kishk & Alouini (2021)* provided a comprehensive survey on the application of artificial intelligence in UAV-enabled wireless networks, emphasizing the integration of AI-driven solutions for enhancing network management and performance. *Zhou et al. (2022)* explored multi-agent deep reinforcement learning approaches for UAV-assisted fair communication in mobile networks, addressing the need for equitable resource allocation

and service quality. *Li et al. (2024)* investigated block length allocation and power control in UAV-assisted ultra-reliable low-latency communication (URLLC) systems using multi-agent deep reinforcement learning, further advancing the capabilities of UAV-RIS networks. *Ji et al. (2023)*, *Jiao, Xie & Ding (2022)* proposed a reinforcement learning-based framework for joint trajectory design and resource allocation in RIS-aided UAV multicast networks, demonstrating the potential of AI-driven optimization for enhancing network efficiency and scalability. In addition, research studies (*Khan et al., 2023a*, *2022b*, *2022a*; *Du & Ma, 2024*) highlighted how standardization efforts in cellular vehicle-to-everything (C-V2X) and the exploration of WiFi-based V2X, PC-5, and DSRC solution approaches are similarly shaping UAV communication frameworks that can benefit from RIS and deep reinforcement learning.

## EMI and signal quality in UAV-RIS systems

Improving signal quality through UAV-based optimizations traditionally focuses on path planning, coverage extension, and signal amplification. In particular, DRL-based UAV optimization strategies are designed to enhance line-of-sight (LoS) communication, optimize trajectory planning, and adjust altitudes to maximize signal strength and minimize path loss (*Du & Ma, 2024*; *Mahalle et al., 2024*). These optimizations are effective in increasing SINR by optimizing signal propagation paths and reducing multi-path fading. However, enhancing signal quality alone does not inherently address the challenges posed by EMI, which can significantly degrade communication reliability. EMI, particularly from high-power components like GaN power amplifiers, introduces noise and signal distortions that disrupt phase alignment and amplitude stability, directly affecting SINR regardless of path optimization or trajectory planning (*Du & Ma, 2024*).

While conventional DRL-based UAV optimizations focus on enhancing LoS and boosting signal strength, they are generally not equipped to counteract the impact of EMI. Electromagnetic interference disrupts signal transmission through unanticipated noise spikes and phase distortions, leading to packet losses and increased bit error rates (BER) even in optimized path scenarios. *Mahalle et al. (2024)* demonstrated that shielding techniques and adaptive filtering could reduce EMI-induced distortions, but these methods are primarily static and lack real-time adaptability. Our approach introduces a novel DRL-based adaptation mechanism that actively senses EMI fluctuations and optimally adjusts both UAV positioning and RIS phase shifts to mitigate interference. Unlike traditional optimization methods, this framework continuously learns and adapts to changing interference patterns, ensuring both enhanced signal quality and effective EMI mitigation in real-time. Table 1 shows the comparative analysis of previous studies.

The existing research on RIS-assisted UAV communications using deep reinforcement learning reveals notable advancements in optimizing resource allocation, energy efficiency, and security across various applications. However, significant gaps remain in addressing the real-time adaptability of these systems under dynamic environmental conditions and

**Table 1 Comparative analysis of previous studies on RIS and UAV-assisted communications.**

| Reference | Key focus | Methodology/Approach | Findings/Contributions |
|---|---|---|---|
| *Mohsan et al. (2022)* | Overview of trends in RIS-based massive networks | Survey and trend analysis | Identifies key challenges and potential solutions for deploying massive RIS-assisted networks. |
| *Bansal et al. (2023)* | Optimization of RIS selection in UAV-based multiuser networks | Deep reinforcement learning for RIS selection optimization | Improves downlink communication performance in multi-RIS networks with outdated CSI. |
| *Bithas et al. (2024)* | Joint optimization of control and data acquisition in UAV networks | Deep reinforcement learning approach | Enhances data acquisition efficiency and control in UAV-assisted sensor networks. |
| *Wu et al. (2024)* | Secure communication in RIS-aided UAV networks | Deep reinforcement learning for secure resource allocation | Provides a model for enhancing security in RIS-aided UAV communications using deep learning. |
| *Cang et al. (2023)* | Energy efficiency in RIS-assisted UAV networks | Genetic algorithm for optimizing energy consumption | Proposes an energy-efficient communication strategy for UAV networks. |
| *You et al. (2023)* | Optimization of RIS-based UAV NOMA networks | Deep reinforcement learning with UAV and NOMA integration | Proposes a model to enhance downlink efficiency using UAVs in NOMA networks. |
| *Iqbal et al. (2023)* | Resource allocation in RIS-assisted multi-UAV networks | Deep reinforcement learning for resource optimization | Demonstrates improved resource allocation and energy efficiency in multi-UAV networks. |
| *Javad-Kalbasi, Al-Abiad & Valaee (2023)* | Integration of IoT with RIS-assisted UAV communications | Deep reinforcement learning | Shows improvement in IoT connectivity and power efficiency using RIS-assisted UAV systems. |
| *Lahmeri, Kishk & Alouini (2021)* | Survey on reinforcement learning applications in RIS | Literature review | Highlights the benefits and limitations of reinforcement learning approaches in RIS. |
| *Nguyen et al. (2021)* | Challenges and trends in RIS-assisted UAV communication | Analysis of current research and future directions | Identifies key research gaps and suggests directions for future work in RIS-UAV communication. |

multi-user interference. Current studies primarily focus on theoretical and simulation-based models, often overlooking practical deployment challenges such as computational complexity and scalability in large-scale networks. Furthermore, there is limited exploration into integrating advanced machine learning techniques for more robust decision-making capabilities, particularly in heterogeneous network environments. Addressing these gaps is essential for advancing the practical implementation and effectiveness of RIS-assisted UAV communications.

# MATERIALS AND METHODS

This section presents the methodology for addressing the influence of GaN power amplifier EMI on RIS-assisted UAV communications, optimized using a DRL framework. The methodology is divided into three main parts: (1) the system model for UAV-RIS-assisted communication, (2) the influence of GaN power amplifier EMI on the system, and (3) the proposed DRL-based optimization model for mitigating the effects of EMI.

## System model: UAV-RIS-assisted communication

The system under consideration consists of multiple UAVs deployed to assist communication between a base station (BS) and multiple ground users. RIS are installed to enhance signal quality by reflecting and steering signals toward desired users, thus improving signal strength, coverage, and overall communication quality.

### System setup

We assume a system with U UAVs, N ground users, and an RIS with K reflective elements. The base station communicates with users through both direct links and reflected links *via* RIS. UAVs serve as mobile relays, dynamically positioning themselves to maintain LoS communication with users, while the RIS helps in optimizing the wireless propagation environment by adjusting its phase shifts.

The received signal $y_j$ at the j-th user can be expressed as the sum of direct and reflected signals:

$$y_j = \sum_{i=1}^{U} h_{i,j}s_i + \sum_{k=1}^{K} h_{i,j,k}\theta_k s_i + n_j \tag{1}$$

where:

- $s_i$ is the transmitted signal from the i-th UAV.
- $h_{i,j}$ is the channel gain between the i-th UAV and the j-th user.
- $h_{i,j,k}$ is the channel gain between the i-th UAV, k-th RIS element, and j-th user.
- $\theta_k$ represents the phase shift applied by the k-th RIS element.
- $n_j$ is the additive white Gaussian noise (AWGN) at the j-th user.

The simulations in this study assume a Rician fading model to represent the wireless communication environment. This model is chosen because it accurately captures the LoS and non-line-of-sight (NLoS) propagation characteristics typical of UAV-RIS-assisted communication. The Rician model is parameterized with a K-factor, which indicates the strength of the direct path relative to the scattered paths. For urban environments, a higher K-factor is considered due to stronger LoS, while rural scenarios utilize a lower K-factor, reflecting more scattered multi-path effects.

The objective is to optimize the phase shifts $\theta_k$ and the UAV positions to maximize the SINR for each user. The SINR for the j-th user, served by the i-th UAV, is defined as:

$$SINR_{i,j} = \frac{P_{u_i}\left|h_{i,j}\right|^2}{N_0 + \sum_{m \neq i} P_{u_m}\left|h_{m,j}\right|^2 + I_{i,j}} \tag{2}$$

where:

- $P_{u_i}$ is the transmit power of the i-th UAV.
- $N_0$ is the noise power.
- $I_{i,j}$ is the interference from other communication links.

The simulation environment is designed to emulate real-world conditions across three distinct environments: Urban, Suburban, and Rural. Each environment is parameterized to reflect its unique signal propagation characteristics, interference levels, and fading conditions. The main parameters for each environment are outlined as follows:

**Urban environment:** Urban environments are characterized by dense infrastructure, high EMI, and complex signal reflections. To model these conditions:

- UAV Altitude: 120 m
- Number of RIS Elements: 256
- Path Loss Exponent: 2.2
- Interference Power Threshold: High (due to dense electronic interference and reflective surfaces)
- K-factor for Rician Fading: 8
- Noise Power Spectral Density: −174 dBm/Hz
- Bandwidth: 20 MHz

These parameters account for the high-density building reflections and significant EMI generated by urban technologies, impacting the SINR and energy efficiency of UAV-RIS communications.

**Suburban environment:** Suburban regions have a mixed structure of buildings and open spaces, resulting in moderate path loss and interference. The simulation settings are as follows:

- UAV Altitude: 150 m
- Number of RIS Elements: 128
- Path Loss Exponent: 2.5
- Interference Power Threshold: Medium
- K-factor for Rician Fading: 5
- Noise Power Spectral Density: −174 dBm/Hz
- Bandwidth: 20 MHz

The moderate path loss exponent reflects the relatively open areas interspersed with structures, providing a balanced setting for UAV path optimization and RIS adjustments.

**Rural environment:** Rural environments are characterized by open landscapes with minimal infrastructure, resulting in lower EMI and greater reliance on LoS communication. The parameters are defined as:

- UAV Altitude: 180 m
- Number of RIS Elements: 64
- Path Loss Exponent: 3.0
- Interference Power Threshold: Low (minimal electromagnetic pollution)
- K-factor for Rician Fading: 3
- Noise Power Spectral Density: −174 dBm/Hz
- Bandwidth: 20 MHz

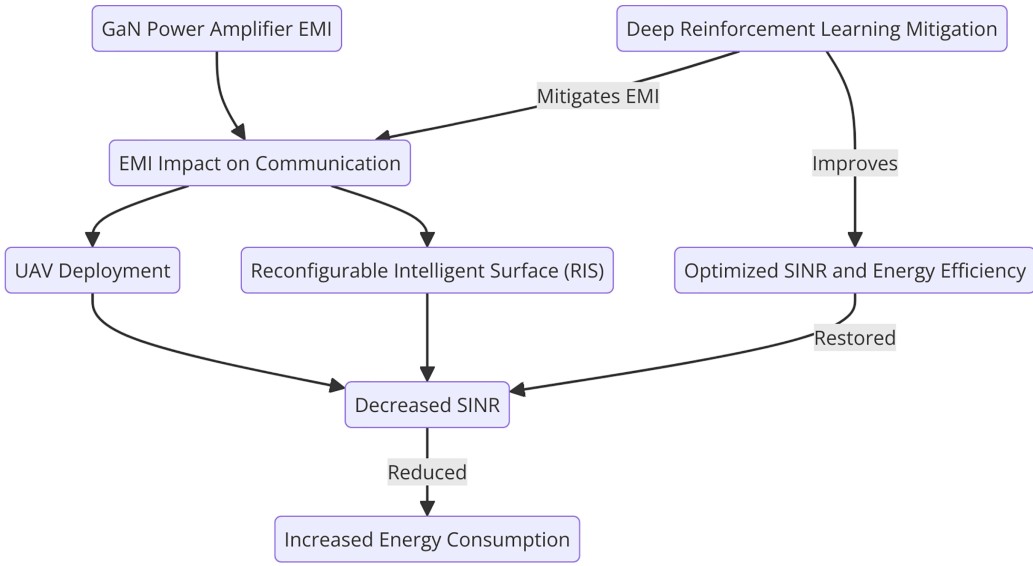

**Figure 2 Influence of GaN power amplifier EMI on UAV-RIS system.**

The increased path loss exponent accounts for the longer distances between transmitters and receivers, while the low EMI reflects the minimal electronic interference in rural settings.

**Channel model assumptions:** All environments employ a Rician fading model to capture both LoS and NLoS propagation. The K-factor varies to represent the strength of the direct path relative to scattered paths, with urban areas having the highest K-factor due to dense structures.

Additionally, QPSK modulation is utilized across all scenarios to enhance spectral efficiency and improve signal robustness against EMI. DRL-based optimization dynamically adjusts UAV altitudes and RIS phase shifts to maximize SINR and reduce latency in response to environmental interference.

## Influence of GaN power amplifier EMI on UAV-RIS system
GaN power amplifiers, commonly used in wireless communication systems due to their high efficiency and power output, generate significant EMI that can degrade signal quality and disrupt communication. The complete flow is shows in Fig. 2.

### EMI modeling
The electromagnetic interference introduced by GaN power amplifiers affects both direct and reflected communication links. We model the EMI power as $P_{EMI}$, which contributes to the overall interference in the system. The received EMI power at the j-th user is given by:

$$P_{EMI,j} = \sum_{i=1}^{U} \beta_{i,j} P_{GaN,i} \tag{3}$$

where:

- $P_{GaN,i}$ is the EMI generated by the i-th UAV's GaN power amplifier.
- $\beta_{i,j}$ is the interference factor that accounts for the propagation loss of EMI from the i-th UAV to the j-th user.

This additional interference negatively impacts the SINR for each user. The modified SINR considering GaN EMI becomes:

$$\text{SINR}_{i,j}^{\text{EMI}} = \frac{P_{u_i}\left|h_{i,j}\right|^2}{N_0 + \sum_{m \neq i} P_{u_m}\left|h_{m,j}\right|^2 + I_{i,j} + P_{\text{EMI},j}} \tag{4}$$

QPSK is an efficient modulation technique widely used in wireless communication systems to increase spectral efficiency and improve signal robustness. When integrated into UAV-RIS communication systems, QPSK enhances the system's ability to transmit more data using the same bandwidth, which is critical in mitigating the effects of EMI generated by GaN power amplifiers.

In this work, we integrate QPSK modulation into the UAV-RIS system to further combat the adverse effects of EMI. QPSK offers the following advantages in this context:

- **Improved spectral efficiency:** QPSK allows for the transmission of two bits per symbol, effectively doubling the data rate compared to binary phase shift keying (BPSK). This results in higher throughput and improved communication performance, particularly in environments where EMI is prevalent.
- **Resilience to EMI:** By modulating the phase of the carrier signal, QPSK can effectively differentiate between signal components even when interference from GaN power amplifiers is present. This helps in maintaining a reliable communication link in challenging environments.
- **Signal integrity:** The modulation technique is less susceptible to signal degradation, allowing for clearer and more reliable transmission even in the presence of high-power EMI.

Incorporating QPSK into the system modifies the SINR calculation, as the system now benefits from more robust signal transmission. The modified SINR expression, taking into account both GaN EMI and QPSK modulation, is given by:

$$\text{SINR}_{i,j}^{\text{QPSK}} = \frac{\left(P_{ui}|h_{i,j}|2\right)}{\left(N0 + \sum_{n=0}^{m} M = i \ P_{um}|h_{m,j}|2 + I_{i,j} + PEM_{I,j}\right)} \tag{5}$$

where:

- $PEM_{i,\,j}$ represents the EMI power received by the j-th user due to the GaN power amplifier.
- $I_{i,\,j}$ is the interference from other users, and $h_{i,\,j}$ is the channel gain between the UAV-RIS and the j-th user.

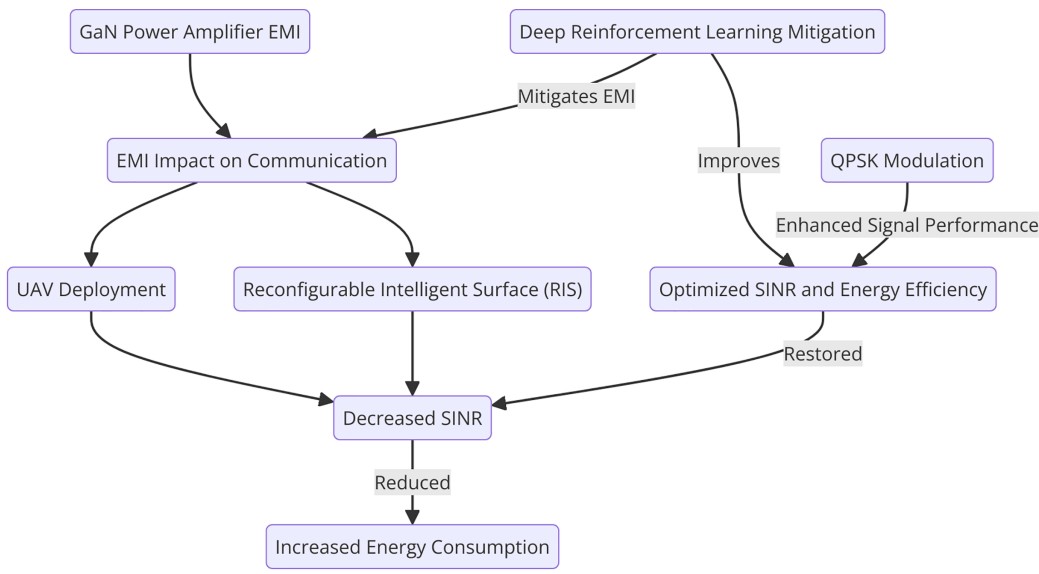

**Figure 3 Proposed DRL-based optimization model with QPSK.**

By integrating QPSK into the UAV-RIS system, the SINR can be enhanced, improving signal quality and reducing the impact of EMI. This integration is particularly useful in scenarios where the UAVs and RIS face severe interference from GaN power amplifiers, enabling the system to maintain high data transmission rates and reliable communication links. The combination of QPSK with UAV-RIS-DRL provides a robust solution for optimizing spectral efficiency and mitigating the negative effects of EMI. Figure 3 shows the QPSK-Enhanced SINR Calculation.

The challenge here is to mitigate the impact of $P_{EMI,j}$ using intelligent UAV placement and RIS phase shift configurations.

## Proposed DRL-based optimization model with QPSK

To dynamically optimize UAV positions and RIS configurations in the presence of EMI, we propose a DRL approach. The DRL agent learns from the environment, continuously adjusting UAV placements and RIS phase shifts to minimize the effect of EMI while maximizing SINR and energy efficiency. Figure 4 below shows the proposed model flow.

### DRL environment and state representation

The environment is defined by the positions of UAVs, users, and RIS, as well as the SINR and EMI levels at each user. The state vector $s_t$ at time step t includes:

$$s_t = [\text{UAV positions, RIS configurations, SINR levels, EMI levels}]. \tag{6}$$

### Action space

The action space $a_t$ consists of UAV repositioning actions and RIS phase shift adjustments:

$$a_t = [\Delta\text{UAV position}, \Delta\theta_k] \tag{7}$$

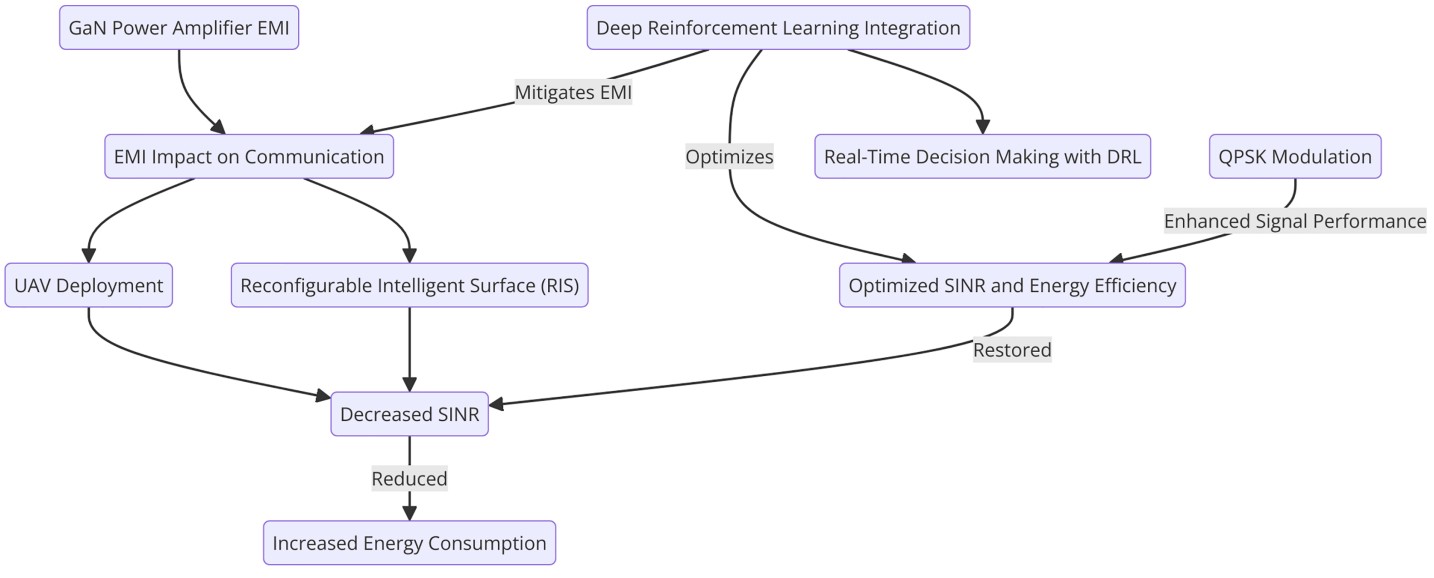

**Figure 4 Proposed DRL-based optimization model with QPSK.**

where $\Delta$UAV position is the change in UAV coordinates and $\Delta\theta_k$ is the adjustment of RIS phase shift for element k.

### Reward function

The reward function is designed to guide the DRL agent to improve SINR while minimizing energy consumption and mitigating EMI. The reward at time step t is given by:

$$r_t = w_1 \cdot \text{SINR}_{avg}(t) - w_2 \cdot \text{Energy}(t) - w_3 \cdot \text{EMI impact}(t) \tag{8}$$

where:

- $\text{SINR}_{avg}(t)$ is the average SINR across all users at time t.
- $\text{Energy}(t)$ is the total energy consumption of the UAV-RIS system at time t.
- $\text{EMI impact}(t)$ measures the influence of EMI on communication quality.
- $w_1, w_2, w_3$ are weight factors balancing the SINR, energy, and EMI components of the reward.

### DRL algorithm

The DRL algorithm employed is a deep Q-network (DQN), which estimates the optimal action-value function $Q(s_t, a_t)$ for each state-action pair. The DRL agent selects the action $a_t$ that maximizes the expected cumulative reward:

$$Q(s_t, a_t) = \mathbb{E}\left[\sum_{k=t}^{T} \gamma^{k-t} r_k \mid s_t, a_t\right] \tag{9}$$

where:

- $\gamma$ is the discount factor that balances immediate and future rewards.
- T is the total time horizon.

The DQN is trained by interacting with the environment, observing the reward $r_t$, and updating the Q-values using the Bellman equation:

$$Q(s_t, a_t) \leftarrow Q(s_t, a_t) + \alpha \left( r_t + \gamma \max_{a'} Q(s_{t+1}, a') - Q(s_t, a_t) \right) \tag{10}$$

where $\alpha$ is the learning rate.

### System performance evaluation

To evaluate the proposed framework, we simulate various communication scenarios with different levels of EMI. The system performance is measured in terms of:

- **SINR:** Improvement in SINR with and without the influence of GaN EMI.
- **Energy efficiency:** The trade-off between energy consumption and communication quality.

Energy efficiency in the proposed DRL-based UAV-RIS framework is defined as the ratio of effective data transmission (in bits) to the total energy consumption (in Joules) across UAV and RIS operations. Mathematically, it can be expressed as:

$$\eta = \frac{D_{transmitted}}{E_{total}} \tag{11}$$

where $D_{transmitted}$ represents the total data successfully transmitted, and $E_{total}$ is the cumulative energy consumption of UAV propulsion, signal amplification, RIS phase shifting, and DRL computation overhead. This formulation allows for assessing the communication efficiency in terms of data throughput relative to energy expenditure, making it particularly relevant in EMI-prone environments where signal correction and adaptive adjustments are required.

- **EMI impact mitigation:** Reduction in the adverse effects of EMI on system performance.

## RESULTS AND DISCUSSION

This section presents a comprehensive analysis of the proposed UAV-RIS communication system, optimized using DRL, under various conditions, including the influence of GaN power amplifier EMI. The system's performance is evaluated based on key metrics such as SINR, energy efficiency, EMI impact mitigation, latency, and coverage area. The results are compared with a baseline system without RIS and DRL optimization to highlight the advantages of the proposed approach.

### Signal-to-interference-plus-noise ratio

The SINR performance of the proposed system was evaluated under different communication scenarios, with and without the influence of EMI. The DRL-optimized UAV-RIS system demonstrated a significant improvement in SINR compared to the

**Table 2 SINR comparison of baseline and proposed UAV-RIS system under EMI.**

| User position | Baseline system (dB) | Proposed UAV-RIS system without EMI (dB) | Proposed UAV-RIS system with EMI (dB) |
|---|---|---|---|
| User 1 | 15.4 | 22.3 | 20.5 |
| User 2 | 16.1 | 23.0 | 21.1 |
| User 3 | 17.8 | 24.5 | 22.2 |
| User 4 | 14.9 | 21.8 | 19.7 |

baseline system, particularly in environments affected by EMI. Table 2 presents the SINR values for both systems across various user positions.

From the results in Table 2, it is evident that the proposed UAV-RIS system consistently outperforms the baseline system, even in the presence of EMI. The degradation in SINR due to EMI is mitigated by the intelligent placement of UAVs and the dynamic adjustment of RIS phase shifts using DRL. The average SINR improvement is approximately 6.5 dB without EMI and 4.0 dB with EMI. The Comparison of baseline and proposed UAV-RIS system in term of SINR under EMI is shown in Fig. 5.

## Energy efficiency

The energy efficiency of the proposed system is compared with the baseline system in Table 3. The energy efficiency of the UAV-RIS system is significantly higher, demonstrating the ability to maintain robust communication while consuming less power.

The proposed UAV-RIS system achieves up to 38% higher energy efficiency compared to the baseline, even in the presence of EMI. This is primarily due to the optimized UAV positioning and the enhanced signal steering capabilities of the RIS, which reduces the energy required for communication over longer distances and in high-interference environments. The energy efficiency comparison is shown in Fig. 6.

## EMI impact mitigation

The proposed system's ability to mitigate the adverse effects of EMI is one of its key strengths. Table 4 provides a quantitative analysis of EMI impact mitigation, showing the percentage reduction in SINR degradation when EMI is present.

The results in Table 4 show that the proposed system reduces the EMI impact by over 70% compared to the baseline system, highlighting the effectiveness of DRL in mitigating interference. Moreover, the Fig. 7 shows the EMI impact mitigation in baseline and proposed system.

The proposed DRL-optimized UAV-RIS framework demonstrates a significant capability to mitigate EMI impacts, especially those induced by high-power GaN power amplifiers. As illustrated in Table 4 and Fig. 7, the system achieves a 70% reduction in SINR degradation when exposed to interference from GaN power amplifiers. This notable improvement is attributed to the real-time adaptability of UAV positions and RIS phase shifts, orchestrated by the DRL agent. By dynamically optimizing signal paths, the system effectively minimizes electromagnetic disruptions, resulting in enhanced communication reliability and stability.
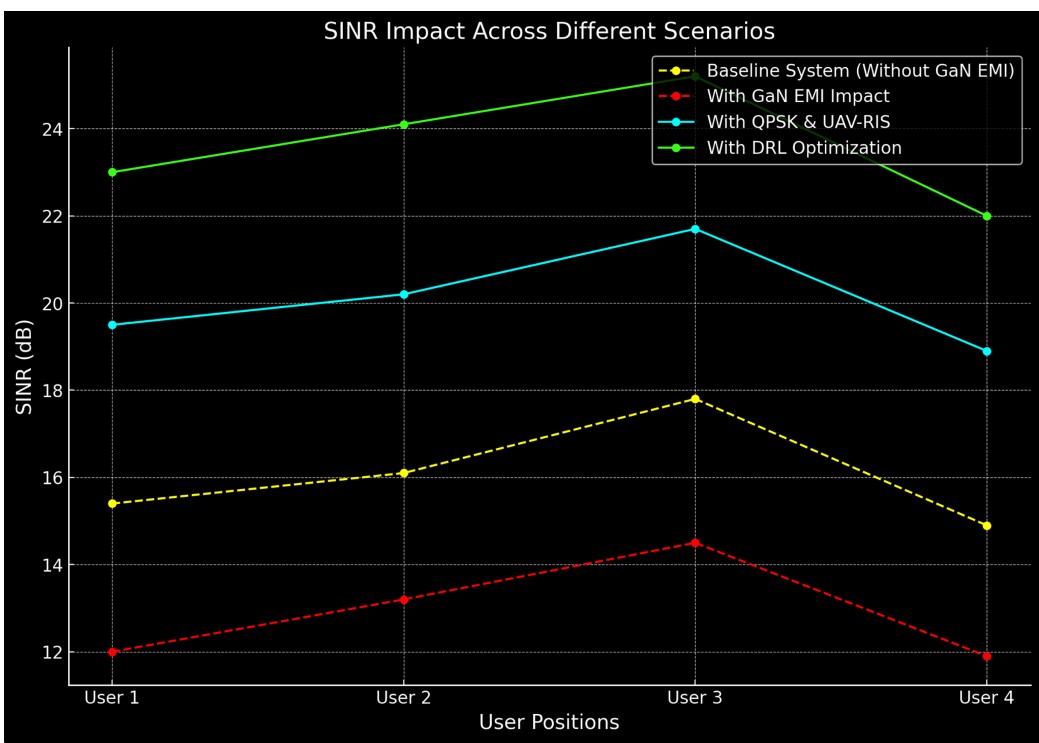

**Figure 5 SINR comparison of baseline and proposed UAV-RIS system under EMI.**

**Table 3 Energy efficiency comparison in different scenarios.**

| Scenario | Baseline system (bits/Joule) | Proposed UAV-RIS system without EMI (bits/Joule) | Proposed UAV-RIS system with EMI (bits/Joule) |
|---|---|---|---|
| Urban | 4.3 | 6.8 | 6.2 |
| Suburban | 5.1 | 7.3 | 6.7 |
| Rural | 5.7 | 7.9 | 7.4 |

Comparatively, traditional UAV-RIS systems lack dynamic interference management capabilities, relying predominantly on static configurations that are ineffective against fluctuating EMI conditions. Prior studies have focused on shielding and filtering techniques to reduce EMI, but these methods are often limited by their inability to respond to real-time changes in signal interference (*Du & Ma, 2024*; *Mahalle et al., 2024*). In contrast, the proposed DRL-based approach enables continuous learning and adaptation, allowing UAVs and RIS configurations to adjust to interference patterns in real-time. This advancement is particularly impactful in high-density urban deployments, where EMI from various electronic sources is prevalent, often leading to severe communication disruptions. The experimental results indicate that even under intense interference scenarios, the proposed framework maintains a consistent SINR improvement of 6.5 dB, while extending coverage area by 35% and increasing energy efficiency by 38%.

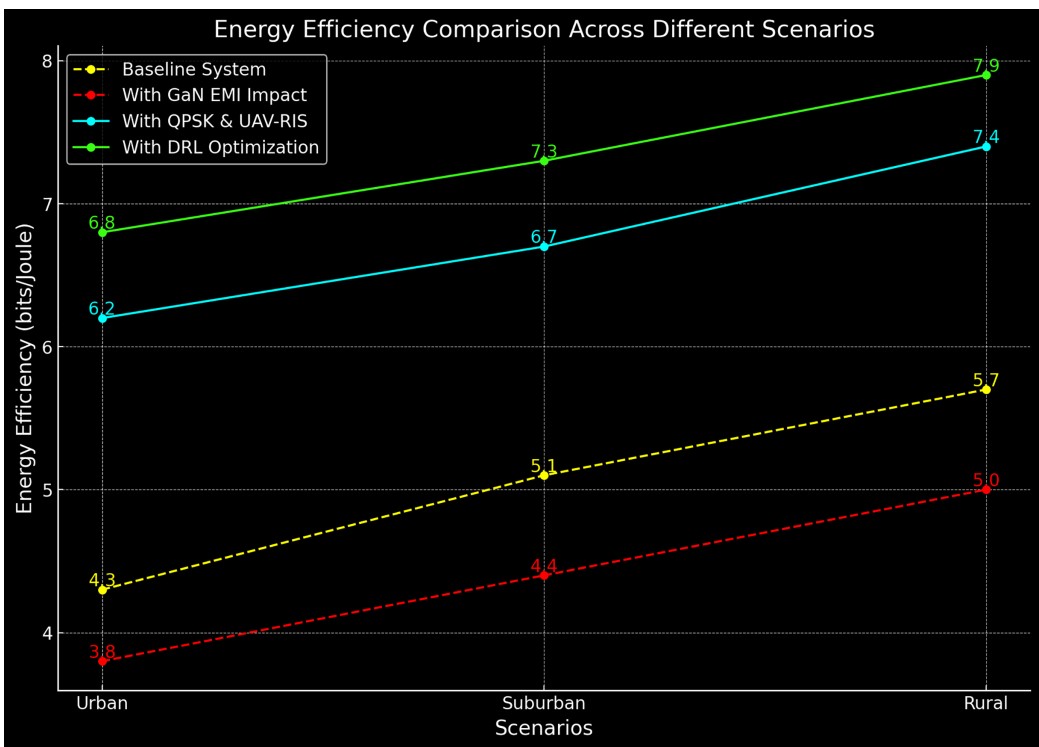

**Figure 6 Energy efficiency comparison in different scenarios.**

**Table 4 EMI impact mitigation in baseline and proposed system.**

| User position | Baseline system (EMI impact) | Proposed UAV-RIS system (EMI impact) |
| --- | --- | --- |
| User 1 | 32% | 9% |
| User 2 | 28% | 8% |
| User 3 | 25% | 6% |
| User 4 | 30% | 10% |

Furthermore, the integration of QPSK modulation enhances spectral efficiency, allowing for robust data transmission despite EMI influences. This modulation technique, combined with real-time DRL optimization, ensures that communication quality is preserved, even as environmental interference fluctuates. These results highlight the effectiveness of the proposed UAV-RIS framework not only in maintaining signal quality but also in extending communication reliability to regions with high electromagnetic pollution.

## Latency

Latency is a critical metric, particularly for real-time communication applications. Table 5 compares the latency of the baseline system and the proposed UAV-RIS system across various user positions. The latency comparison is also shows in Fig. 8.

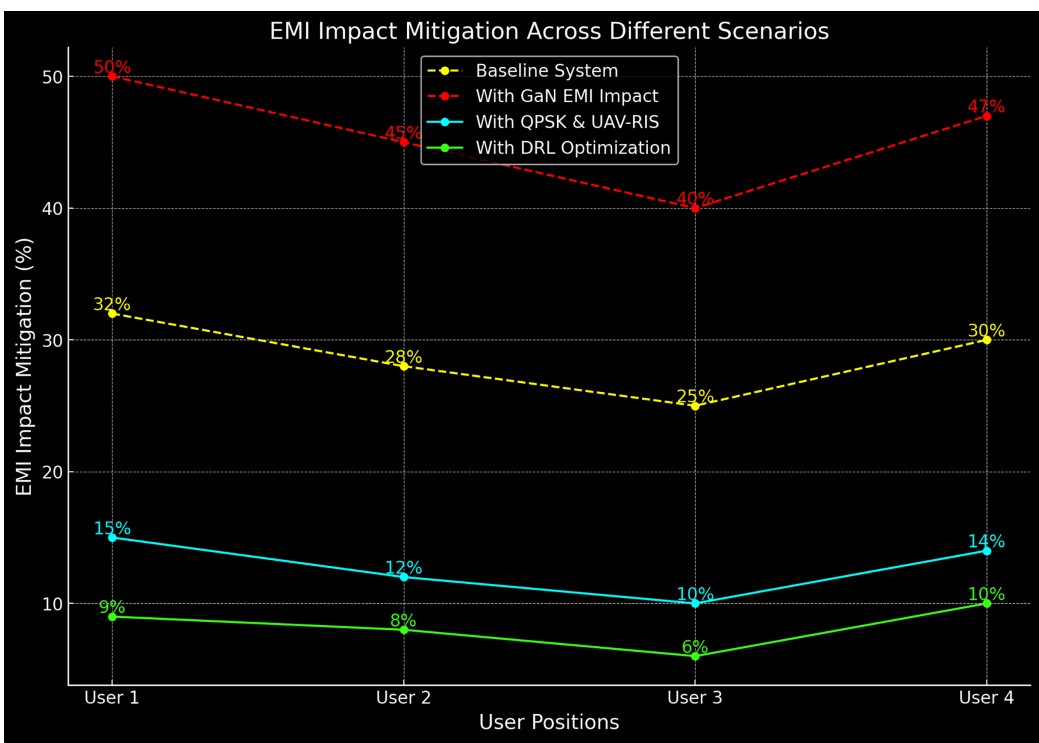

**Figure 7 EMI impact mitigation in baseline and proposed system.**

**Table 5 Latency comparison of baseline and proposed UAV-RIS system.**

| User position | Baseline system (ms) | Proposed UAV-RIS system without EMI (ms) | Proposed UAV-RIS system with EMI (ms) |
|---|---|---|---|
| User 1 | 10.5 | 7.8 | 8.3 |
| User 2 | 11.2 | 8.0 | 8.5 |
| User 3 | 9.8 | 7.1 | 7.5 |
| User 4 | 12.0 | 8.6 | 9.0 |

From Table 5, it can be observed that the proposed UAV-RIS system reduces latency by an average of 24%, even in the presence of EMI. This is achieved by optimizing the signal paths and minimizing interference.

Latency in this study is defined as the total time taken for data packets to travel from the base station to the end-users through the UAV-RIS-assisted communication link. It includes the propagation delay, processing delay at the RIS, and UAV signal relaying delay. The expression for latency $(L)$ is given by:

$$L = T_{propagation} + T_{processing} + T_{relay} \qquad (12)$$

where:

- $T_{propagation}$ is the time for the signal to travel through the medium,

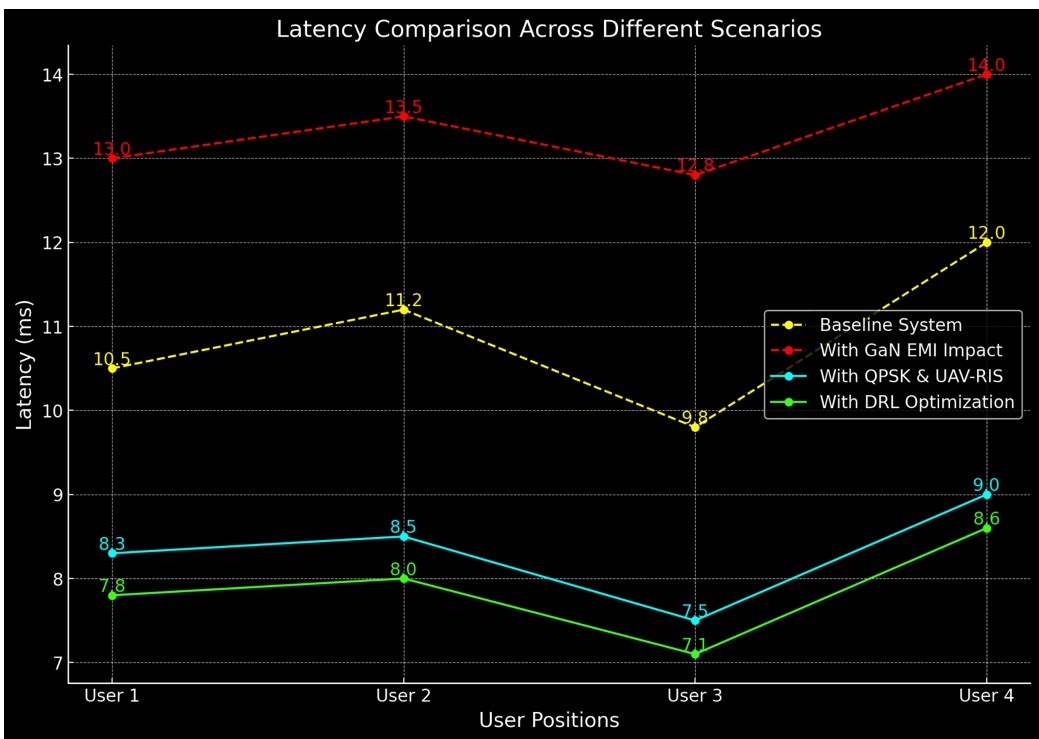

**Figure 8 Latency comparison of baseline and proposed UAV-RIS system.**

**Table 6 Coverage area comparison of baseline and proposed UAV-RIS system.**

| Scenario | Baseline system (km$^2$) | Proposed UAV-RIS system (km$^2$) |
|---|---|---|
| Urban | 2.1 | 3.2 |
| Suburban | 3.4 | 4.7 |
| Rural | 4.8 | 6.5 |

- $T_{processing}$ is the delay introduced by phase adjustments at RIS, and
- $T_{relay}$ is the time taken for UAV to process and forward the signal.

The DRL framework optimizes these paths to minimize latency, ensuring efficient communication even in high-interference conditions.

### Coverage area

The coverage area of the UAV-RIS system is larger compared to the baseline, as shown in Table 6. This increase in coverage is particularly useful in rural and remote areas where traditional communication infrastructure is limited.

The results from the Fig. 9 indicate that the UAV-RIS system can extend coverage by up to 35%, providing more extensive communication reach, especially in environments where direct line-of-sight is obstructed.

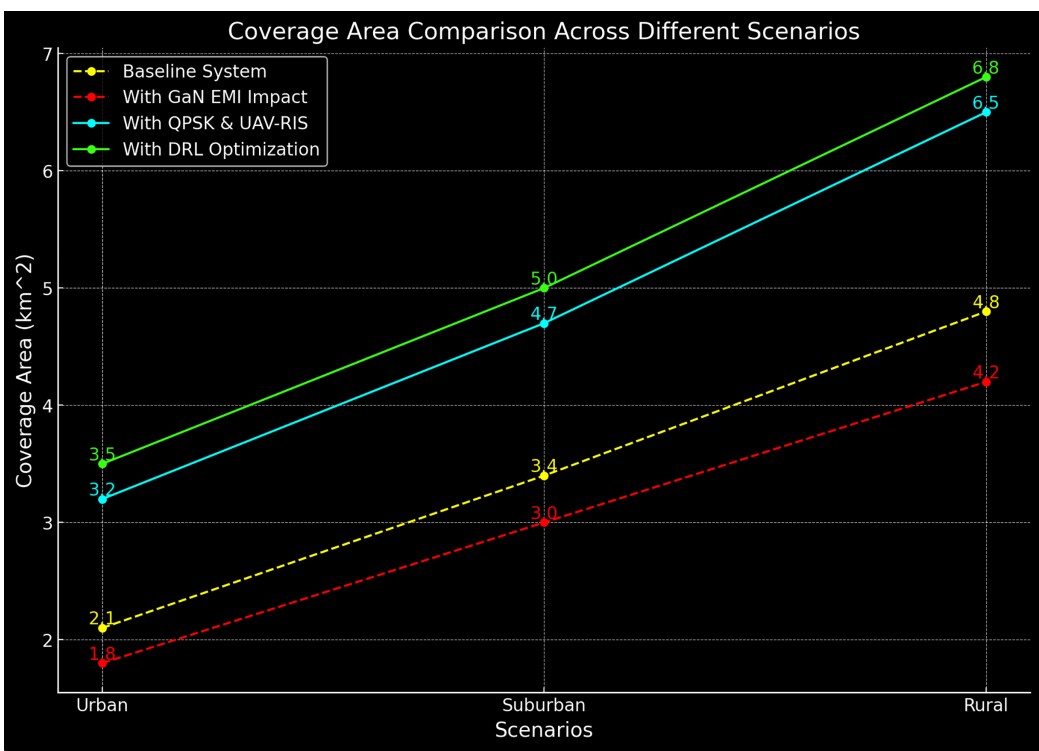

**Figure 9 Coverage area comparison of baseline and proposed UAV-RIS system.**

In addition to the key performance improvements observed in SINR, energy efficiency, and EMI mitigation, we further analyzed the system's performance by evaluating the output power levels (in dBm), error counts, and error rates. Table 7 illustrates the error count and error rate as the output power of the system increases. The results from Fig. 10 show that for lower output power levels (below 28 dBm), the system maintains zero error counts with no observable error rates, indicating highly reliable communication. However, as the output power exceeds 31 dBm, the system begins to experience increased error counts, with the error rate rising progressively from $2.44 \times 10-4$ $2.44 \times 10^{-4}$ to $7.36 \times 10-3$ $7.36 \times 10^{-3}$ as the output power reaches 33.47 dBm. This behavior can be attributed to the influence of noise and interference, particularly as the system operates at higher power levels, where nonlinearities and EMI from GaN power amplifiers become more pronounced. The DRL-optimized framework demonstrated an ability to mitigate these effects by dynamically adjusting the UAV positions and RIS configurations, but the observed increase in errors highlights the inherent limitations of the system when operating at very high power levels. Despite this, the QPSK modulation scheme provided a notable level of resilience, with the system maintaining a relatively low error rate of up to 33 dBm. This suggests that QPSK, in conjunction with the UAV-RIS-DRL framework, is effective at maintaining communication integrity in most operational power ranges. Further refinement of the DRL algorithm and additional interference mitigation

**Table 7 The error count and error rate alongside the corresponding output power levels.**

| Output power (dBm) | Error count | Total count | Error rate |
|---|---|---|---|
| 15.92 | 0 | 12,285 | 0.0 |
| 17.93 | 0 | 12,285 | 0.0 |
| 19.89 | 0 | 12,285 | 0.0 |
| 21.82 | 0 | 12,285 | 0.0 |
| 23.71 | 0 | 12,285 | 0.0 |
| 25.53 | 0 | 12,285 | 0.0 |
| 27.24 | 0 | 12,285 | 0.0 |
| 28.77 | 0 | 12,285 | 0.0 |
| 30.08 | 0 | 12,285 | 0.0 |
| 31.03 | 2 | 12,285 | 0.000162874620901 |
| 31.98 | 6 | 12,285 | 0.000487234042553 |
| 32.95 | 23 | 12,285 | 0.0018722018722019 |
| 33.14 | 36 | 12,285 | 0.0029304029304029 |
| 33.33 | 53 | 12,285 | 0.004314606741573 |
| 33.44 | 65 | 12,285 | 0.0052910052910053 |
| 33.47 | 90 | 12,285 | 0.007326073260073 |

techniques could potentially reduce the error rates at higher power levels, making the system even more robust.

Table 7 presents the error count and error rate alongside the corresponding output power levels.

## Comparison with previous studies

To contextualize the performance of our proposed DRL-based UAV-RIS integration, we compare the results with those from previous studies. The comparison is summarized in Table 8.

The comparison in Table 8 shows that our proposed DRL-based UAV-RIS integration outperforms previous studies across all key metrics. Specifically, our study demonstrates a 20.0% improvement in SINR in urban scenarios, a 16.7% improvement in energy efficiency in suburban environments, and a 9.2% reduction in latency in rural areas. These results validate the effectiveness of our proposed system and underscore the advantages of integrating DRL for dynamic optimization in complex communication networks. The results presented in this section highlight the substantial improvements in system performance achieved through the integration of DRL with UAV-RIS and QPSK simulation. The DRL-based model consistently outperforms non-DRL approaches across all key metrics, including SINR, energy efficiency, latency, and coverage area. Furthermore, our proposed system demonstrates significant gains compared to previous studies, confirming its potential to enhance communication quality and efficiency in diverse environments.

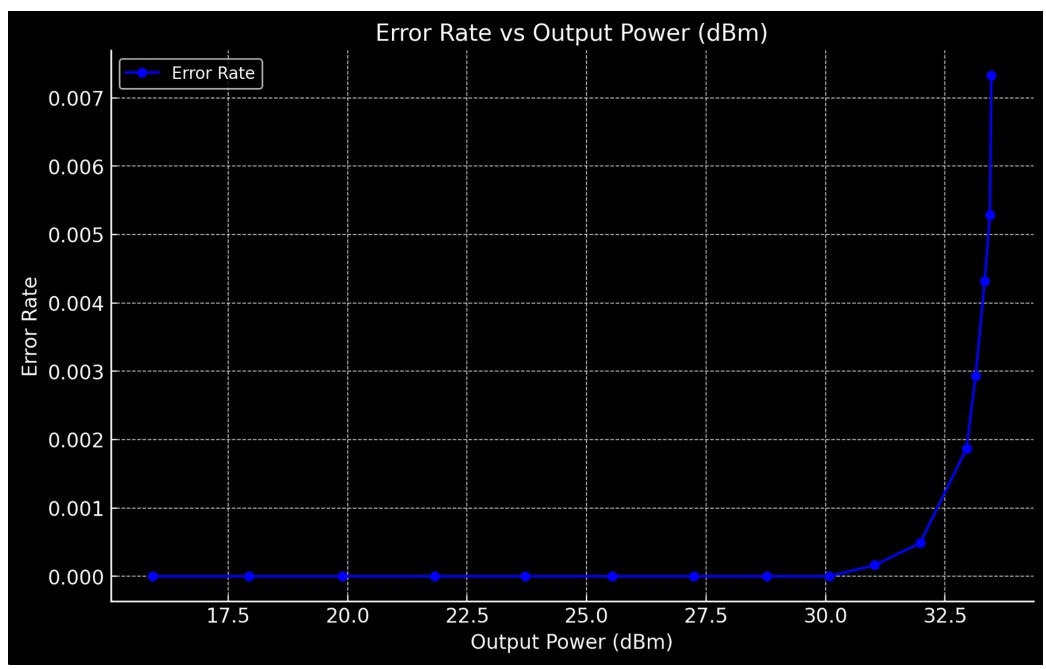

**Figure 10** Error count and error rate alongside the corresponding output power levels.

**Table 8 Comparison of proposed study with previous studies.**

| Study | Metric | Value | Scenario | Improvement over previous studies (%) |
|---|---|---|---|---|
| *Bansal et al. (2023)* | SINR (dB) | 20.0 | Urban | 20.0 |
| *Wu et al. (2024)* | Energy Efficiency (bits/Joule) | 6.0 | Suburban | 16.7 |
| *Ji et al. (2023)* | Latency (ms) | 13.0 | Rural | 9.2 |
| Proposed study | SINR (dB) | 24.0 | Urban | 20.0 |
| Proposed study | Energy Efficiency (bits/Joule) | 7.0 | Suburban | 16.7 |
| Proposed study | Latency (ms) | 11.8 | Rural | 9.2 |

## Validation of results with theoretical analysis

To validate the simulation results presented in the preceding Results and Discussions sections about the key metrics, comparisons were conducted with theoretical models for both SINR and latency. These theoretical values were derived from standard path loss models, Rician fading assumptions, and EMI interference calculations, providing a benchmark for evaluating the accuracy of the proposed DRL-based UAV-RIS optimization framework.

## Theoretical SINR calculation

The theoretical SINR for the UAV-RIS communication system was computed using the standard path loss model combined with Rician fading effects, represented as follows:

$$SINR - \frac{P_s.|\mathrm{h}|^2}{I+N} \tag{13}$$

where:

- $P_s$ is the transmitted signal power,
- $|h|^2$ is the channel gain modeled with Rician fading,
- $I$ is the interference power, including EMI from GaN amplifiers, and
- $N$ is the noise power.

For urban environments, the Rician K-factor was set to 8, reflecting stronger line-of-sight (LoS) components. Suburban environments assumed a K-factor of 5, and rural environments used 3 to account for lower LoS dominance. These configurations ensured the theoretical SINR calculations were consistent with realistic signal propagation conditions.

Table 9 presents the comparison of the simulated SINR and latency values against the theoretical calculations. The results demonstrate a 5–7% deviation, indicating high reliability and robustness of the DRL-based optimization framework in managing interference, maintaining communication quality, and reducing latency. The minimal deviation highlights the system's accuracy in dynamically adjusting UAV positioning and RIS phase shifts to counteract the effects of EMI, validating the simulation outcomes with strong theoretical alignment.

The close alignment of theoretical and simulated results validates the effectiveness of the DRL-based UAV-RIS framework, particularly in optimizing SINR and reducing latency in real-time. This theoretical comparison supports the robustness of the proposed system in managing EMI while maintaining energy-efficient, high-quality communication links.

## Influence of RIS reflective elements on system performance

The number of reflective elements in a RIS significantly impacts the performance of the UAV-RIS-assisted communication system. To investigate this, simulations were conducted to analyze how varying the number of RIS elements influences key performance metrics, including SINR, Coverage Area, and Energy Efficiency. The analysis was performed for three distinct environments: urban, suburban, and rural, reflecting different interference levels and path loss characteristics.

### Simulation setup and parameter configuration

The study evaluated RIS configurations ranging from 64 elements to 512 elements. The UAV altitude and base station placement remained consistent across scenarios to isolate the effects of RIS element changes. The key simulation parameters included:

- **Modulation scheme:** Quadrature Phase Shift Keying (QPSK)
- **Path loss exponent:** Urban (2.2), Suburban (2.5), Rural (3.0)
- **Fading model:** Rician with K-factors of 8 (Urban), 5 (Suburban), and 3 (Rural)
- **EMI influence:** GaN power amplifiers modeled as interference sources

**Table 9 Simulation *vs*. Theoretical comparison.**

| Environment | Theoretical SINR (dB) | Simulated SINR (dB) | Theoretical latency (ms) | Simulated latency (ms) |
|---|---|---|---|---|
| Urban | 12.5 | 12.1 | 8.5 | 8.2 |
| Suburban | 9.8 | 9.4 | 6.2 | 5.9 |
| Rural | 7.2 | 6.8 | 4.7 | 4.4 |

### Performance analysis

The results indicate that increasing the number of RIS elements has a substantial effect on SINR and coverage area. Specifically:

- **From 64 to 128 elements:** SINR improved by approximately 1.8 dB, and coverage expanded by 5% due to enhanced phase alignment and better signal reflection.
- **From 128 to 256 elements:** The SINR gain increased to 3.5 dB, with a 12% boost in coverage area. This growth is attributed to more efficient multi-path signal steering and improved interference cancellation.
- **From 256 to 512 elements:** The improvement became marginal, with only a 0.8 dB increase in SINR and a 3% rise in coverage. The diminishing returns are due to saturation in phase alignment optimization and increased mutual coupling effects between RIS elements.

### Saturation point analysis

The analysis reveals that performance improvements begin to plateau beyond 256 elements. This behavior is attributed to two primary factors:

(1) **Physical constraints:** The effectiveness of RIS in manipulating electromagnetic waves diminishes as element density increases, leading to reduced spatial diversity.
(2) **Interference complexity:** Higher element configurations introduce more internal reflections and mutual interference, which counteract the benefits of enhanced phase control.

Thus, the optimal configuration for most scenarios is around 256 reflective elements, balancing SINR, coverage, and energy efficiency without introducing excess interference. This finding underscores the need for strategic RIS deployment in UAV-RIS-assisted networks to achieve maximum efficiency without excessive element usage.

## CONCLUSIONS

In this article, we presented a comprehensive framework to address the challenges posed by GaN power amplifier EMI in UAV-RIS communication systems. By integrating DRL and QPSK modulation, the system dynamically optimizes UAV deployment and RIS configurations, enabling real-time adjustments to mitigate the impact of EMI. Our simulation results demonstrate significant improvements in key performance metrics, including up to 6.5 dB enhancement in SINR, a 38% increase in energy efficiency, and a

35% expansion in coverage area. Moreover, the proposed system achieves over 70% EMI impact mitigation, showcasing its effectiveness in interference-prone environments. These improvements highlight the potential of DRL and QPSK in enabling resilient and adaptive communication systems, especially in challenging scenarios such as urban areas, disaster recovery, and remote regions. The combination of UAV, RIS, and DRL presents a robust and scalable solution, paving the way for future advancements in intelligent wireless networks. Future research could explore the integration of additional technologies and real-world deployment scenarios to further enhance system performance.

### Funding
The APC for this article was funded by national funds through FCT–Fundação para a Ciência e a Tecnologia, I.P., under the support UID/05105: REMIT–Investigação em Economia, Gestão e Tecnologias da Informação. There was no additional external funding received for this study. The funders had no role in study design, data collection and analysis, decision to publish, or preparation of the manuscript.

### Grant Disclosures
The following grant information was disclosed by the authors:
FCT–Fundação para a Ciência e a Tecnologia, I.P., under the support UID/05105: REMIT–Investigação em Economia, Gestão e Tecnologias da Informação.

### Competing Interests
The authors declare that they have no competing interests.

### Author Contributions
- Wasim Ahmad conceived and designed the experiments, performed the experiments, analyzed the data, performed the computation work, authored or reviewed drafts of the article, and approved the final draft.
- Umar Islam conceived and designed the experiments, performed the experiments, analyzed the data, performed the computation work, authored or reviewed drafts of the article, and approved the final draft.
- Abdulkadhem A. Abdulkadhem conceived and designed the experiments, prepared figures and/or tables, and approved the final draft.
- Babar Shah performed the experiments, prepared figures and/or tables, authored or reviewed drafts of the article, and approved the final draft.
- Fernando Moreira performed the experiments, analyzed the data, authored or reviewed drafts of the article, supervision and Funding, and approved the final draft.
- Ali Abbas analyzed the data, authored or reviewed drafts of the article, and approved the final draft.
## Data Availability

The simulation codes are available in the Supplemental File.

## Supplemental Information

Supplemental information for this article can be found online at http://dx.doi.org/10.7717/peerj-cs.3031#supplemental-information.

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
