# Peer review of "Enhancing reliable and energy-efficient UAV communications with RIS and deep reinforcement learning"

_PeerJ Computer Science, doi:10.7717/peerj-cs.3031_

## Round 0.1 · original submission · Major Revisions

Dear authors,

You are advised to critically respond to all comments point by point when preparing an updated version of the manuscript and while preparing for the rebuttal letter. Please address all comments/suggestions provided by reviewers, considering that these should be added to the new version of the manuscript.

Kind regards,
PCoelho

Reviewer 1 ·

Basic reporting

The article was easy to read and understand
The literature review was okay, however, it didn't cover the EMI issues. Hence, it is suggested to include some description about the EMI impact and issues since the article focuses on those topics.

Recheck on the fig 1. Is Fig. 1 representing the conceptual UAV_RIS-assisted system? As the notation of Fig. 1 stated as previous UAV RIS system, does it being referred from existing works? If it is adopted from any reference, please include/ mention it.

Experimental design

The introduction highlights the aims of the study. However, it didn't highlight the main aims of the study.
What is the main objective of the work?
- Investigating the influence of GaN power amplifier EMI on the RIS-assisted UAV communication systems OR
- propose integrating DRL with UAV-RIS systems to provide a dynamic and adaptive solution that continuously learns from the environment

Quadrature Phase Shift Keying (QPSK) modulation, which is an important element of the work, was not mentioned in the objective part of the introduction.

Methodology is okay

Validity of the findings

As the article highlighted that its contribution is on the following:
a) A novel DRL-based framework that dynamically adapts UAV positions and RIS configurations to mitigate the adverse effects of GaN power amplifier EMI on communication performance.
b) An in-depth analysis of the influence of EMI on RIS-assisted UAV communication systems, and how DRL can be used to optimize signal quality and energy efficiency in real-time.
c) Comprehensive simulation results show significant improvements in SINR, energy efficiency, coverage, and latency in EMI-prone environments when using the proposed DRL-based approach

The author should highlight and comprehensive discussion about the results that have been obtained. How does it reflect on mitigating against the EMI, and what are the differences from previous work?

Additional comments

What do you mean by this: UAVs using DRL focused on improving signal quality and coverage, but did not account for the challenges posed by EMI? (under introduction section)

Doesn’t improving the signal quality also mean eliminating the EMI effect? Please elaborate further.

Reviewer 2 ·

Basic reporting

In this paper, a DRL-based framework is introduced to dynamically adjust UAV positions and RIS configurations, mitigating GaN power amplifier EMI effects. It analyzes EMI's impact on RIS-assisted UAV communication and demonstrates how DRL optimizes signal quality and energy efficiency. Simulations show significant improvements in SINR, energy efficiency, coverage, and latency. The reference list that is included is complete, while the paper is well written, and the results provided could be helpful for people working in this area.

Experimental design

The research questions are satisfactory. However, a few points should be clarified.
1. It is not clear how energy efficiency is defined
2. It is not clear which fading model has been assumed for obtaining the simulation results
3. It is not clear how the latency has been defined
4. More details regarding the simulation setup should be included, for example, what are the exact parameters assumed in the urban, suburban, and rural environments

Validity of the findings

The simulation results have not been compared with theoretical ones, so the authors should elaborate more on the validation of the results presented.

Additional comments

The authors should investigate how the number of RIS reflected elements influences the performance of the system under consideration

---

## Round 0.2 · accepted · Accept

Dear authors, we are pleased to verify that you meet the reviewer's valuable feedback to improve your research.

Thank you for considering PeerJ Computer Science and submitting your work.

Kind regards
PCoelho

Reviewer 2 ·

Basic reporting

The authors have satisfactory addressed the comments raised in the previous round of review. The paper can be accepted as is

Experimental design

It is ok

Validity of the findings

It is ok

Additional comments

It is ok